# Residual Infusion Performance Evaluation (RIPE): A Single-Center Evaluation of Residual Volume Post-Intravenous Eravacycline Infusion

**DOI:** 10.3390/pharmacy11020075

**Published:** 2023-04-13

**Authors:** Alysa J. Baumann, Kerry O. Cleveland, Michael S. Gelfand, Nicholson B. Perkins III, Angela D. Covington, Athena L. V. Hobbs

**Affiliations:** 1St. Jude Children’s Research Hospital, Memphis, TN 38105, USA; 2University of Tennessee Health Science Center, College of Medicine, Memphis, TN 38163, USA; 3University of Tennessee Health Science Center, College of Pharmacy, Memphis, TN 38103, USA; 4Methodist Le Bonheur Healthcare, Memphis, TN 38104, USA; 5Cardinal Health Innovative Delivery Solutions, Dublin, OH 43017, USA

**Keywords:** pharmacy, antimicrobial stewardship, infectious diseases, pharmacist

## Abstract

Intravenous (IV) drugs are administered through infusion pumps and IV administration sets for patients who are seen in healthcare settings. There are multiple areas of the medication administration process that can influence the amount of a drug a patient receives. For example, IV administration sets that deliver a drug from an infusion bag to a patient vary in terms of length and bore. In addition, fluid manufacturers report that the total acceptable volume range for a 250 mL bag of normal saline can be anywhere from 265 to 285 mL. At the institution chosen for our study, each 50 mg vial of eravacycline is reconstituted using 5 mL of diluent, and the total dose is administered as a 250 mL admixture. This single-center, retrospective, quasi-experimental study evaluated the residual medication volume after the completion of an IV eravacycline infusion in patients admitted during the pre-intervention study period compared to those in the post-intervention study period. The primary outcome of the study was to compare the residual antibiotic volume remaining in the bags following IV infusions of eravacycline before and after the implementation of interventions. The secondary outcomes included the following: comparing the amount of the drug lost in the pre- and post-intervention periods, determining whether the amount of residual volume was affected by nursing shifts (day versus night), and lastly assessing the cost of facility drug waste. On average, approximately 15% of the total bag volume was not infused during the pre-intervention period, which was reduced to less than 5% in the post-intervention period. Clinically, the average estimated amount of eravacycline discarded decreased from 13.5 mg to 4.7 mg in the pre- and post-intervention periods, respectively. Following the statistically significant results of this study, the interventions were expanded at this facility to include all admixed antimicrobials. Further studies are needed to determine the potential clinical impact when patients do not receive complete antibiotic infusions.

## 1. Introduction

Patients admitted to healthcare systems often require intravenous (IV) medications for various indications. Nursing staff members are taught to program drug infusion pumps with the volume to be infused and the infusion rate. While the infusion rate is ordered by the physician, the volume to be infused is determined by the pharmacy based on the drug and primary base solution volume. With the exception of restrictive volume patient situations, the total volume to be infused often mimics the primary base solution volume (e.g., 50 mL, 100 mL, 250 mL). However, this can pose a problem as IV fluid manufacturers report the total acceptable volume range to be 265–285 mL for a 250 mL sodium chloride 0.9% infusion bag [1]. This overfill volume is not included in the volume to be infused and therefore not reflected in pump programming. The overfill volume along with the residual volume, in theory, could contribute to a significant amount of a given drug that does not reach the patient. However, published research in the literature documenting this theory regarding residual drug volume is absent.

The Infusion Therapy Standards of Practice published by the Infusion Nursing Society (INS) in January 2021 dedicated a section to medication infusion and administration. Specifically, the INS highlights the potential significant loss of medication, noting up to 35% can be lost due to residual volume within the administration set. However, this statement only applies to 50 and 100 mL solutions. To minimize residual drug loss, the document recommends administering intermittent infusions as a secondary infusion or flushing the administration set with the primary solution. Interestingly, the INS recommends ensuring antimicrobials are infused with a minimal loss of drugs as a component of antimicrobial stewardship. Again, these recommendations only apply to small-volume infusions, defined as having a volume less than or equal to 100 mL [2].

Additionally, the Institute for Safe Medication Practices (ISMP) warns that if the same tubing is used later for medication or fluid administration, the residual volume left in the tubing could result in an inadvertent bolus of the medication. The ISMP recommends minimizing the small-volume infusion residual volume in tubing with various strategies, such as using microbore tubing and flushing the tubing after the infusion [3]. The ISMP’s recommendations address the residual volume in administration set tubing but fail to recognize the other residual volume locations.

Acknowledging both the INS’s and ISMP’s statements, several smart infusion pump vendors have built forcing functions. For example, pumps may require small-volume intermittent infusions to be administered via a secondary infusion. A secondary infusion method automatically switches to a primary infusion after the completion of the small-volume infusion. This ensures the entire small-volume infusion is administered while also flushing the tubing using the primary infusion solution.

While the topic of residual volume is addressed, the comments fail to include large-volume infusions and its multiple contributing factors. This could, in part, be due to the absence of data in the published literature involving residual volume for infusions larger than 100 mL. The purpose of this study was to quantify the residual medication volume in completed IV infusions greater than 100 mL, formulate interventions, and evaluate the interventions’ impact.

## 2. Materials and Methods

This IRB-approved, single-center, retrospective, quasi-experimental study was conducted at a flagship 617-bed, urban, academic adult facility. Patients were identified using an electronic medical record (EMR) report and were included if they were 18 years or older and received at least one dose of intravenous eravacycline during their admission. Eravacycline was selected for evaluation due to its infusion volume, frequency of use, and color. Eravacycline has a large infusion volume (250 mL) that has not been evaluated in the current literature. Its moderate use in the facility also showed that this drug could provide enough doses to evaluate it without presenting so many doses that they could not all feasibly be collected, as would be the case for a more commonly used antimicrobial. Lastly, its bright yellow color when admixed had originally alerted the infectious diseases physicians that residual drug volume in large-volume admixtures after completion of infusions was a potential concern in the facility.

Patients were excluded if they were pregnant or had acute coronavirus (COVID-19) infections. The pre-intervention study period was 1 July through 30 July 2021, whereas the post-intervention study period was 1 November through 30 November 2021. The intervention period was 1 August 2021 through 31 October 2021.

Interventions included updating the volume to be infused on each patient’s specific eravacycline label to reflect the clinically significant diluent. This included a calculated infusion rate which reflected the maximum possible volume per dose with the addition of the clinically significant diluent. Of note, each 50 mg eravacycline vial was diluted with 5 mL of sterile water for injection as the diluent. Additionally, education was provided to nursing colleagues to encourage the administration of the entire drug volume and completion of the IV infusion. The patient-specific labels were updated to include a comment in the administration section to “INFUSE AT CURRENT RATE UNTIL BAG EMPTY” to encourage nursing staff to adhere to the updated protocol of infusing the total drug volume.

This institution uses 297 cm standard bore tubing, which can hold up to 25 mL of fluid [4]. Completed infusion bags were collected at least once daily, seven days a week. The primary investigator manually measured the residual volume of eravacycline remaining in each infusion bag and tubing using 30 mL Luer Lock syringes. Residual volume was removed from infusion bag and tubing using the administration set tubing’s top and bottom ports. Percentage of total infusion volume was calculated using residual volume left in the 250 mL sodium chloride 0.9% carrier fluid. Residual drug dose and percentage of total dose were calculated for each data point based on patient-specific dose and clinically significant diluent volume added to 250 mL of carrier fluid. Actual patient dose and total bag volume were used when calculating residual drug amount, percentage of total dose, and facility waste. For example, a patient dose of 100 mg eravacycline would equal a total bag volume of 260 mL, which includes the 250 mL carrier fluid and 10 mL of clinically significant diluent used to reconstitute two 50 mg eravacycline vials. Cost analysis was calculated using average wholesale price of 50 mg eravacycline vial multiplied by the number of vials needed to admix the patient-specific dose.

This institution utilized twelve-hour shifts for nursing staff, with shift changes occurring at 0700 and 1900. Drug administration times were used to identify whether the administration was performed by day or night shift nursing staff.

The primary outcome of the study was to compare residual antibiotic volume remaining in the infusion bag following administration of IV eravacycline before and after implementation of interventions. The secondary outcomes included comparing the amount of drugs lost in the pre- and post-intervention periods, determining whether amount of residual volume was affected by nursing shifts (day versus night), and lastly assessing the cost of facility drug waste.

## 3. Results

A total of 67 doses among 16 patients were included in this study across both intervention periods. The primary indications for eravacycline included lower respiratory tract infections (*n* = 7) and osteomyelitis (*n* = 3) (Figure 1 and Figure 2). The majority of patients were discharged home or to a nursing facility (*n* = 10, 62.5%), while six (37.5%) study participants expired during their hospital admission.

The pre-intervention period included 46 doses among nine patients while the post-intervention period consisted of 21 doses among seven patients. Their baseline characteristics were similar, with no statistically significant differences between the pre- and post-intervention groups (Table 1). The average length of stay and duration of the eravacycline therapy were also similar between the groups (Table 1). The combined average length of stay was 38.5 days. The average amount of residual volume after a 250 mL eravacycline infusion during the pre-intervention period was 38.0 mL ± 15.6 mL, which is approximately 15% of the total bag volume. The largest volume remaining in the infusion bag during the pre-intervention period was 85 mL, equal to 34% of the total bag volume. The post-intervention average residual antibiotic volume was 12.2 ± 10.1 mL, which is less than 5% of the total bag volume. The difference in the average residual volume between the pre- versus post-intervention groups (Table 2) was statistically significant *p* < 0.0001 (Figure 3). During the pre-intervention period, the collected doses were equally split between the day shift (*n* = 22) and evening shift (*n* = 24). When analyzed by shift, the average residual volume for the day shift was 44.0 ± 16.4 mL, while the average residual volume for the night shift was 32.6 ± 12.9 mL (*p* = 0.01). Our post-intervention shift analysis showed that the average residual antibiotic volume was 12.0 ± 12.0 mL for the day shift (*n* = 11) and 12.5 ± 8.2 mL for the night shift (*n* = 10) (*p* = 0.91). The average remaining amount of eravacycline was 13.5 mg during the pre-intervention period, which equated to 15.3% of the total dose. The average remaining dose of eravacycline was 4.7 mg in the post-intervention period, equating to approximately 4.7% of total dose remaining in the residual volume. The pre- and post-intervention eravacycline doses remaining in the residual volume were statistically significant (*p* < 0.0001).

A cost analysis was completed using THE average wholesale price for the 50 mg eravacycline vials. The percentage of residual volume remaining in the 250 mL sodium chloride 0.9% carrier fluid was multiplied by the quantity of eravacycline vials used to compound the preparation. The residual volume in the pre-intervention period equaled USD 893.45 compared to USD 161.04 in the post-intervention period. This is a source of facility waste equal to USD 10,721.40 annually (Table 2).

## 4. Discussion

Residual drug volume has been established as an important point of concern, but data evaluating large-volume infusions are lacking. Incomplete drug administration can have several implications for the patient, the pharmacy department, and the healthcare system in its entirety. Readmissions, increased lengths of stay, patient statuses, and the development of antimicrobial resistance are all of special interest to healthcare system metrics and patient outcomes. According to a 2018 National Hospital Ambulatory Medical Care Survey distributed by the CDC, 523,000 emergency department visits resulted in hospital admissions with principal hospital discharge diagnoses of infectious and parasitic diseases [5]. This large number of infectious disease admissions presents a potential for residual volume to be an issue for healthcare systems across the country. Although residual volume is not at the forefront of drug administration practices, residual drug volume itself has been mentioned in the ISMP and INS Standards of Practice.

There is a lack of published literature describing the effects of residual volume following large-volume intravenous infusions, but residual drug volume has been mentioned by both the ISMP and INS. The INS stated there was a “significant potential loss of medication with 50- and 100-mL solutions of up to 35% of medication loss due to residual volume in the administration set; greatest percentage loss was with 50-mL volumes” [2]. This is consistent with the results of this study in which the largest residual volume measured during the pre-intervention study period was 34% of the total bag volume. There have been a few studies investigating residual volume, but each has had limitations.

A study conducted by Cooper et al. [6] calculated, rather than measured, residual volume within administration sets. This study found that the mean residual volume was 13.1 mL for gravity administration sets and 16.7 mL for pump administration sets. The study also discovered that up to 21% of the drugs were discarded and ultimately did not reach the patients. However, only 33% of the doses were larger than 100 mL and the residual volume was calculated rather than physically measured. Another study conducted by Bolla et al. [7] evaluated antimicrobial drug loss by using a hypothetical patient scenario. The study found an average of 15.83 mL (with a range of 15.06–16.47 mL) resided within the 235 cm long administration tubing. The study estimated the percentage of drugs lost in the administration set to be 33%. Our study confirms, in a real-world clinical setting, the hypothesis that a significant amount of residual volume can be discarded following an intravenous medication infusion. Additionally, our study found that, on average, 15.3% (with a maximum of 33.3%) of the total eravacycline dose was being discarded during the pre-intervention period. Although our study did not evaluate patient clinical outcomes, the effects of not receiving an entire drug dose raises countless concerns regarding possible repercussions, such as a longer duration of therapy leading to an increased risk of adverse reactions and antimicrobial resistance. In the case of eravacycline specifically, eravacycline expresses non-linear, concentration-dependent activity, which emphasizes the importance of requiring adequate drug concentrations to achieve the disruption of bacterial protein synthesis that results in bacteriostatic activity [8]. If up to a third of the medication is not reaching the patient, there is an inadequate drug concentration necessary for infectious disease antimicrobial therapy.

A further question with clinical implications arose: an average of approximately 10% more drugs reached the patients during the post-intervention period compared to the pre-intervention period. The eravacycline package insert indicated the intravenous administration of eravacycline therapy during phase 1 through phase 3 clinical trials [8]. It is unknown if complete drug administration or residual volume were assessed during the clinical trials. Although this study did not evaluate adverse drug reactions, patients were receiving a lower-than-intended dose during the pre-intervention period, and the post-intervention effects would likely resemble those observed during clinical trials.

A decrease in the residual antibiotic volume was observed following the interventions of education and label modification compared to that of the pre-intervention phase. Following the statistically significant results of this study, the interventions were expanded to all admixture antimicrobials on the healthcare system’s formulary, including acyclovir, amphotericin B, amikacin, ampicillin, ceftolozane/tazobactam, ceftaroline, doxycycline, meropenem, micafungin, nafcillin, sulfamethoxazole/trimethoprim, tobramycin, voriconazole, and piperacillin/tazobactam.

A secondary aim of our study was to determine whether the amount of residual volume was affected by nursing shifts. The average residual volume between the day and night shifts during the post-intervention period was very similar: 12 mL and 12.5 mL, respectively. Interestingly, the average residual volume differed significantly between the day and night shifts during the pre-intervention period: 44.0 ± 16.4 mL and 32.6 ± 12.9 mL respectively (*p* = 0.01). While the exact reason for this difference has yet to be determined, its presence speaks to the variability in nursing practices with regard to drug administration. The standardization of training and drug administration practices could potentially help decrease or eliminate the variability in residual volume and ultimately promote complete drug administration to patients.

This study was not free of limitations, as several doses were discarded entirely, or the bag contents were thrown away by nursing staff members prior to collection. All the residual volumes were measured by one pharmacist for consistency purposes. However, the infusion volume could have been lost during the measuring process through the intravenous tubing, although these volumes would likely be less than 1 mL based on the closed-system method of measurement. Observational bias also could have been a factor following nursing education in the pre-intervention period, as the nursing staff members were asked not to discard the infusion bags after administration completion. However, this observational bias would be consistent between the pre- and post-intervention periods since the nursing staff members were asked to save all the eravacycline infusion bags after the infusions. The difference in the sample size between pre- and post-intervention periods is a statistical limitation of the study. By design, this study was not randomized. Future evaluations of mini-bag or premixed antimicrobials would help determine if residual volume is an issue for more than just admixed drugs. A literature review conducted by Lam et al. [9] specifically analyzed piperacillin-tazobactam residual volume and found that the use of microbore tubing only led to a 1–2% potential dose loss. Of note, piperacillin-tazobactam is a 50 mL or 100 mL infusion [10]. The study recommends increasing infusion volumes to decrease residual volumes. However, our findings repute that recommendation, and we would like to reiterate that residual volume can lead to significant drug loss, even for large-volume infusions. More studies are needed to assess the potential impact of residual antimicrobial volume on clinical outcomes such as readmissions, lengths of stay, clinical improvements, and the development of antimicrobial resistance. In addition, this study deployed three interventions simultaneously, which eliminates our ability to compare the effectiveness of each intervention method.

In summary, the residual volumes following the eravacycline infusions were compared among the pre- versus post-intervention periods. The pre-intervention period had an average residual volume of 38.0 ± 15.6 mL. The three interventions included nursing education, the addition of a clinically significant diluent volume to the total volume to be infused, and an administration instruction stating, “INFUSE AT CURRENT RATE UNTIL BAG EMPTY”. Although this study focused on antimicrobial drug therapy, numerous drugs are administered via intravenous infusions. Further studies are needed to determine the frequency and impact of residual volumes following infusions of other pharmacologic classes.

## Figures and Tables

**Figure 1 pharmacy-11-00075-f001:**
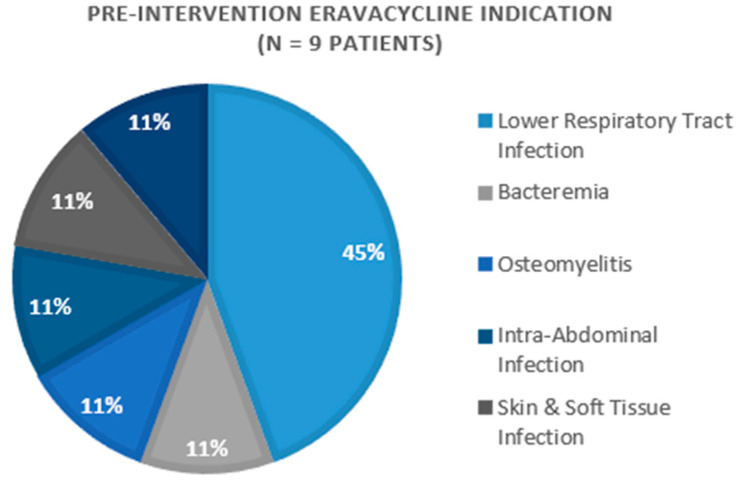
Pre-Intervention Eravacycline Indication (*n* = 9 patients).

**Figure 2 pharmacy-11-00075-f002:**
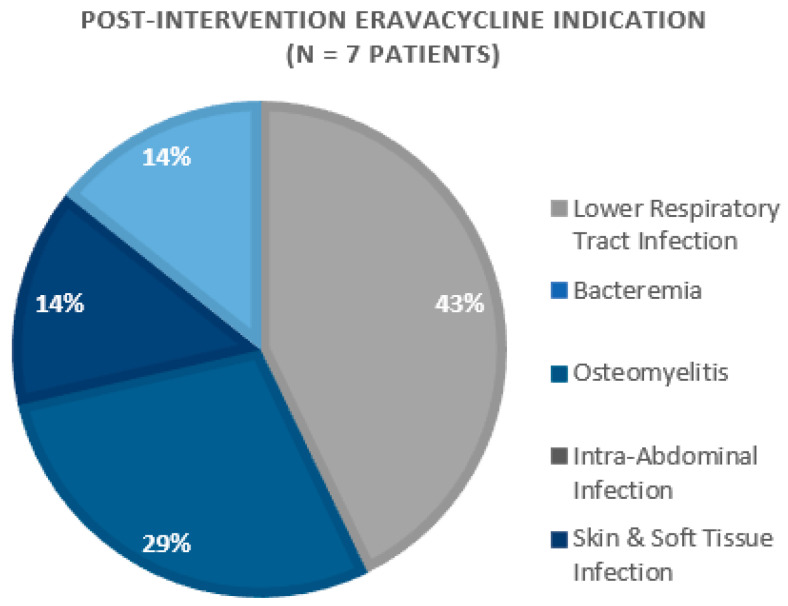
Post-Intervention Eravacycline Indication (*n* = 7 patients).

**Figure 3 pharmacy-11-00075-f003:**
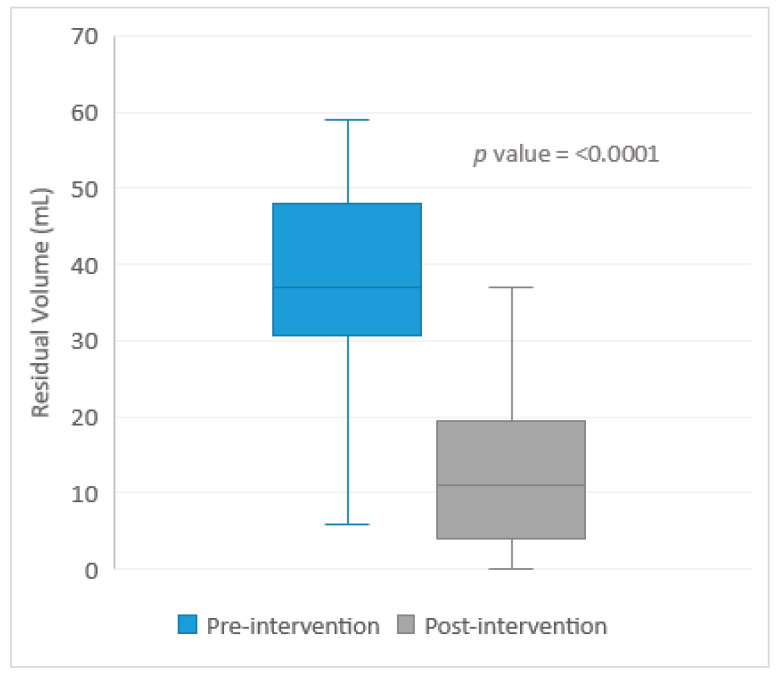
Pre- versus Post-Intervention Residual Volume Comparison.

**Table 1 pharmacy-11-00075-t001:** Baseline Demographics.

Variable	Combined (*n* = 16)	Pre-Intervention (*n* = 9)	Post-Intervention (*n* = 7)	*p* Value (Pre- vs. Post-)
Mean ± S.D. age, yr	56.7 ± 15.7	54.1 ± 16.7	60.0 ± 13.6	0.48
Male, no. (%)	11 (68.8)	5 (55.6)	6 (85.7)	0.20
African American, no. (%)	12 (75.0)	6 (66.7)	6 (85.7)	0.40
Caucasian, no. (%)	3 (18.8)	22 (22.2)	1 (14.3)	0.70
Other ^a^, no. (%)	1 (6.3)	1 (11.1)	0	0.35
Mean ± S.D. weight	87.9 ± 29.0	85.6 ± 22.8	90.8 ± 35.2	0.76
Mean ± S.D. BMI	29.8 ± 10.4	30.0 ± 9.3	29.6 ± 11.7	0.94
Mean ± S.D. Baseline SCr	1.8 ± 1.6	2.1 ± 1.8	1.5 ± 1.2	0.47
Mean ± S.D. doses per patient, no.	5.1 ± 4.0	3.0 ± 2.3	26.5 ± 16.6	0.24
Mean ± S.D. duration of Eravacycline, hours	336.4 ± 461.4	178.7 ± 68.0	267.4 ± 357.6	0.37
Length of stay, days (IQR)	38.5 (1.8–67.7)	40.1 (8.7–106.9)	39.2 (1.8–106.9)	0.93

^a^ Patient self-identified as non-Caucasian or non-African American.

**Table 2 pharmacy-11-00075-t002:** Results.

	Combined (*n* = 16)	Pre-Intervention (*n* = 9)	Post-Intervention (*n* = 7)	*p* Value (Pre- vs. Post-)
Primary Outcome				
Mean ± S.D. remaining volume, mL	30.0 ± 18.4	38.0 ± 15.6	12.2 ± 10.1	<0.0001
Secondary Outcomes				
Range of Remaining Volume, mL	0–85	6–85	0–37	N/A
Day Shift				
Number of doses	33	22	11	N/A
Mean Remaining Volume (mL)	33.3	44.0	12.0	<0.0001
Night Shift				
Number of doses	34	24	10	N/A
Mean Remaining Volume (mL)	26.7	32.6	12.5	<0.0001
Mean ± S.D. Amount of Drug in Residual Volume (mg)	10.76 ± 7.98	13.53 ± 7.49	4.69 ± 5.14	<0.0001
Range of Drug in Residual Volume (mg)	0–40.46	1.63–40.46	0–20.9	N/A
Cost of Facility Waste Study Timeframe Annualized	USD 1054.49 USD 12,653.88	USD 893.45 USD 10,721.40	USD 161.04 USD 1932.48	N/A

## Data Availability

The data presented in this study are available on request from the corresponding author.

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
