# Peer review of "Residual Infusion Performance Evaluation (RIPE): A Single-Center Evaluation of Residual Volume Post-Intravenous Eravacycline Infusion"

_pharmacy, 2023, doi:10.3390/pharmacy11020075_

Round 1

Reviewer 1 Report

I find the topic of this study very interesting.

However, there are a few limitations that have to be addressed and improved:

 1. Research design: as this was not a randomized study and there was no time-control group for the intervention, this has to be addressed in the limitation section of the Discussion.

2.  Sample size: the pre-intervention and post-intervention groups are very different in size. This is very problematic, from a statistical point of view.

3. The finding that day and night shifts have extremely different values in the residual volume of medications, before any interventions, is fascinating and has to be more clearly presented and further addressed in the manuscript.

Kind regards

Reviewer 2 Report

Thank you for your submission.

Some more information on the reconstitution and dilution of the formulation would be helpful for readers (e.g. solutions used); the drug is not even available in my country but I have looked up the US prescribing information.

It is unclear whether the residual volume includes the 5mL of added eravacycline solution e.g. is 85mL actually 34% of the bag volume or is it 33% (if the bag volume is 255mL)?

How exactly was the residual volume physically measured?

“The average remaining amount of eravacycline was 13.5 mg during pre-intervention, which equated to 15.3% of the total dose. The average remaining dose of eravacycline was 4.7 mg in the post-intervention period, equating to approximately 4.7% of total dose remaining in the residual volume.” Again, are these calculations based on the initial bag volume being 250mL or 255mL?

Figure 3 is unnecessary.

The legend for Fig 2 includes a box labelled “indications”.

Line 134. This sentence could be improved: “Similarly, the average length of stay and duration of eravacycline therapy was similar.”

Did what amount to effectively increasing the administered dose of the drug (by about 10-15%) lead to any adverse reactions e.g. palpitations, GI side effects? This leads on to something that perhaps should be discussed (and is more relevant to some of the other antimicrobial drugs, with narrower therapeutic margins, mentioned by the authors): if the clinical trials that established the recommended doses found within the official prescribing information also had the same residual volume issues in practice, aren’t the authors now potentially increasing the dose of all these drugs?
